# Towards Parkinson’s Disease Detection Through Analysis of Everyday Handwriting

**DOI:** 10.3390/diagnostics15030381

**Published:** 2025-02-05

**Authors:** Jeferson David Gallo-Aristizabal, Daniel Escobar-Grisales, Cristian David Ríos-Urrego, Jesús Francisco Vargas-Bonilla, Adolfo M. García, Juan Rafael Orozco-Arroyave

**Affiliations:** 1GITA Lab., Faculty of Engineering, University of Antioquia, Medellín 510010, Colombia; jeferson.gallo@udea.edu.co (J.D.G.-A.); daniel.esobar@udea.edu.co (D.E.-G.); cdavid.rios@udea.edu.co (C.D.R.-U.); jesus.vargas@udea.edu.co (J.F.V.-B.); 2Cognitive Neuroscience Center, Universidad de San Andrés, Buenos Aires B1644BID, Argentina; adolfomartingarcia@gmail.com; 3Global Brain Health Institute (GBHI), University of California San Francisco, San Francisco, CA 94143, USA; 4Trinity College Dublin, D02 R590 Dublin, Ireland; 5Departamento de Lingüística y Literatura, Facultad de Humanidades, Universidad de Santiago de Chile, Santiago 9170020, Chile; 6LME Lab., University of Erlangen, 91054 Erlangen, Germany

**Keywords:** Parkinson’s disease, handwriting, convolutional neural networks, dynamic analysis, natural handwriting tasks

## Abstract

**Background:** Parkinson’s disease (PD) is the second most prevalent neurodegenerative disorder worldwide. People suffering from PD exhibit motor symptoms that affect the control of upper and lower limb movement. Among daily activities that depend on proper upper limb control is the handwriting process, which has been studied in state-of-the-art research, mainly considering non-semantic drawings like spirals, geometric figures, cursive lines, and others. **Objectives:** This paper analyzes the suitability of modeling the handwriting process of digits from 0 to 9 to automatically discriminate between PD patients and healthy control subjects. The main hypothesis is that modeling these numbers allows a more natural evaluation of upper limb control. **Methods:** Two approaches are considered: modeling of the images resulting from the strokes collected by the digital tablet and modeling of the time series yielded by the digital tablet while performing the strokes, i.e., time-dependent signals. The first approach is implemented by fine-tuning a CNN-based architecture, while the second approach is based on hand-crafted features measured upon the time series, namely pressure and kinematic measurements. Features extracted from time-dependent signals are represented following two strategies, one based on statistical functionals and the other one based on creating Gaussian Mixture Models (GMMs). **Results:** The experiments indicate that pressure-based features modeled with functionals are the ones that yield the highest accuracy, indicating that PD-related symptoms are better modeled with dynamic approaches than those based on images. **Conclusions:** The dynamic approach outperformed the image-based model, indicating that the writing process, modeled with signals collected over time, reveals motor symptoms more clearly than images resulting from handwriting. This finding is in line with previous results in the state-of-the-art research and constitutes a step forward to create more accurate and informative methods to detect and monitor PD symptoms.

## 1. Introduction

Parkinson’s disease (PD) is a neurodegenerative disorder that affects the central nervous system, and it is characterized by the progressive loss of dopaminergic neurons in the midbrain [1]. These neurons are mainly responsible for the production of dopamine, which is a neurotransmitter in charge of keeping operational neural pathways associated with mood, motor control, and other functions [2]. Among motor symptoms derived from PD are muscular rigidity, resting tremor, postural instability, and bradykinesia [3]. Motor symptoms negatively impact activities that require highly coordinated movements like handwriting [4]. Anomalies in handwriting include micrographia (abnormal reduction in writing size) and dysgraphia (deficits in graphomotor production) [5]. Other neurodegenerative disorders also affect the handwriting process, for instance, Alzheimer’s disease patients are known for producing abnormal strokes in different handwriting tasks [6,7] Handwriting analyses are carried out via online data or offline data. Online data refer to handwriting signals acquired using digital devices, such as tablets or smart pens; offline data consist of traditional methods like writing with an ink pen on paper. Handwriting using conventional methods could be more natural for elderly patients; however, online data allow dynamic analyses of the handwriting process and enable sensing-relevant bio-signals such as pressure, in-air movements, dynamic horizontal and vertical positions, pen inclinations, and other relevant information [8,9]. For a comprehensive review of methods used to evaluate neurodegenerative disorders considering handwriting biosignals, please see [10].

Online and offline handwriting studies typically consider geometric-based tasks, namely, drawings that involve figures like spirals, meanders, geometric figures, and others [11]. Studies have explored the use of different features including kinematic [12,13,14], geometric, and pressure [15]. One of the main advantages of online handwriting is that it provides the sensor-based signals and also the resulting image, i.e., the drawing. This characteristic has motivated researchers to use image processing methods, mainly based on Convolutional Neural Networks (CNNs), to model those images. For instance, in [16], the authors used time series-based images from drawing tasks. Raw signals collected from the tablet were transformed into matrices to form images which were further processed by a CNN architecture pre-trained on the ImageNet dataset. In another study presented in [17], the authors created spectral representations from tablet signals to model tremor and other symptoms. These signal representations were stacked to form an image and then feed a CNN model with kernel sizes of 1 × 5 and 1 × 3. In [18], authors analyzed drawings from three datasets (PaHaW [19], HandPD [20], and NewHandPD [11]) to increase sample-size. They used an AlexNet model pre-trained on ImageNet and data augmentation techniques. Finally, in [21], authors implemented a CNN with two blocks containing three convolutional layers and one pooling layer. Similarly, in [22], the authors used pre-trained models based on ImageNet to improve the model’s generalization capability. Other datasets like MNIST and UJIpenchars2, which are semantically similar to writing tasks, have also been used to pre-train models [23].

Regarding modeling dynamic information from raw tablet signals, previous studies suggest that dynamic analysis of online handwriting provides suitable information to detect PD motor symptoms such as tremor and bradykinesia [9]. Regarding the offline approach, deep learning architectures mainly based on CNNs have been used to analyze handwritten drawings. On the one hand, the majority of handwriting-related studies consider drawings of geometric figures and spirals. On the other hand, writing tasks that request the patient to write numbers, letters, words, or sentences [19] are more natural and closer to activities of daily living, which is what expert neurologists usually focus on during clinical check-ups. Models focused on these kinds of tasks eventually produce better systems to perform accurate monitoring of patients. Additionally, these semantic writing tasks may have a higher cognitive load to patients, therefore providing information about motor planning (e.g., between-letter or word transitions), which is affected due to lack of coordination in PD [10]. Dynamic signals extracted from writing tasks have been modeled using different dimensions such as kinematic [24,25], spectro-temporal [26], and non-linear [27].

State-of-the-art research shows that deep learning architectures (mainly based on CNN architectures) have been used to model images resulting either from offline handwriting images or from images created by considering online signals. Most studies with promising results are based on drawing tasks [22,28,29]. However, these approaches are not as natural as those based on regular writing tasks, i.e., writing of letters, numbers, words, or sentences, making those models not suitable for performing non-intrusive monitoring of the neurological state of patients.

This paper introduces and compares two different approaches based on online handwriting signals collected from PD patients and healthy controls (HCs). One approach considers resulting images of the tablet handwriting to feed a CNN architecture pre-trained with adapted samples of the MNIST corpus and fine-tuned with Parkinson’s data. The second approach consists of considering online signals and extracting different features, including those related to the pressure and movements of the pen (namely, kinematic). The resulting features are compressed and represented with two statistical approaches: statistical functionals and Gaussian Mixture Models (GMMs). Statistical representations are used as input to a Support Vector Machine (SVM) classifier, the parameters of which are optimized following a k-fold cross-validation strategy. Results show that image-based features are outperformed by dynamic analyses with the pressure feature set using statistical functionals. This likely indicates that vertical control of hand movement is more challenging for PD patients.

The rest of the paper is organized as follows. Section 2 introduces the data considered for this study, Section 3 explains the methodology followed in the paper, Section 4 shows experiments and results, Section 5 elaborates a discussion about findings reported in the study, and finally, Section 6 presents the conclusions derived from this work.

## 2. Data Acquisition and Participants

Our handwriting database (Hw-DB) consists of 104 participants, including 51 PD patients and 53 HC subjects. Forty-seven of the patients were evaluated by an expert neurologist according to the MDS-UPDRS-III scale. The remaining four patients could not come to the clinic for their neurological evaluation because they live in the countryside and were not available for these clinical screenings. Subjects were asked to write the numbers from 0 to 9 using a Wacom Cintiq 13 HD tablet, with a sampling frequency of 180 Hz. This tablet provides six signals: *x* and *y* position, *z* (distance from the tablet surface to the pen tip), azimuth and altitude angles, and pen pressure. Additional details about the participants are shown in Table 1.

## 3. Methodology

Figure 1 summarizes the methodology addressed in this paper. The information from the sequence of numbers is analyzed following two approaches: image-based and dynamic. The first approach only uses the data of the image generated by the position information of the tablet, making this task similar to offline handwriting. This approach considers a Convolutional Neural Network (CNN) pre-trained using sequences created with numbers from the MNIST database. Sizes of digits in the MNIST corpus are changed to artificially create the effect of micrographia. Then, the CNN model is fine-tuned using the handwriting digits of PD and HC subjects. The second approach constitutes a dynamic analysis where we use all information available when data are captured using a digital tablet. Two sets of classical handwriting features are computed from the 6 time-series signals available in online handwriting (*x* and *y* position, *z* (distance from the tablet surface to the pen tip), azimuth and altitude angles, and pen pressure). Finally, feature vectors obtained from the image-based and dynamic analyses are used independently to train two Support Vector Machine (SVM) classifiers to discriminate between PD patients and HC subjects.

### 3.1. Feature Extraction

#### 3.1.1. Image-Based Analysis

This model is based on a CNN architecture, a widely used modeling strategy in image processing [30]. In each layer, different filters are trained to detect patterns while the spatial dimensions of the image are reduced to obtain a compressed image representation. The CNN model training requires many samples to generalize patterns and avoid model overfitting; this represents a limitation in the study of medical data, where only a few samples are available. Transfer Learning (TL) and Data Augmentation (DA) techniques have emerged to address the problem of data scarcity [28,31,32,33].

In this study, we created 7000 synthetic sequences of numbers from 0 to 9 using the digits of the MNIST database [34]. Half of the sequences contain concatenated digits, while the rest are modified to model micrographia, which is a distinct symptom of handwriting in PD patients. The idea behind this approach is to pre-train the architecture to recognize non-modified sequences (synthetic samples of HC subjects, i.e., without micrographia) vs. modified sequences (synthetic samples of PD patients, i.e., with micrographia), which implicitly means having an architecture properly trained to detect micrographia. The starting digit for the modified sequence is randomly chosen from 0 to 5. Once the first modified digit is chosen, the rest of the sequence, until 9, is progressively reduced until the last digit (9) results in half of its original size. Examples of the modified and non-modified sequences of digits from MNIST are shown in Figure 2. In addition, given the fact that several sequences in the Hw-DB exhibited slight rotations resulting from the writing process of participants, all the synthetic sequences (modified and non-modified) were randomly rotated at angles of $-15^{{}^{\circ}},-5^{{}^{\circ}},0^{{}^{\circ}},5^{{}^{\circ}},15^{{}^{\circ}}$. From the set of generated sequences, a subset of 6000 synthetic sequences was used to train a CNN architecture to classify between modified and non-modified sequences. The remaining synthetic sequences were used as a test set. The CNN model employed in this experiment is based on a LeNet architecture because it constitutes a simple model appropriate for the size of the corpus that we could access to develop this study. Additionally, this architecture showed as good results as in a previous work where we modeled digits [35]. This network consists of 3 convolutional layers with 16, 8, and 4 filters, employing a (5×5) kernel. Max-pooling layers were used to reduce the size of each feature map by half. The output feature map was flattened to feed three fully connected layers with 512, 64, and 2 neurons, respectively. The first two fully connected layers employed a rectified linear unit (ReLU) activation function, and the last one made the final decision using a Softmax activation function. The shape of input images is 144 in height and 216 in width.

The pre-trained architecture and its weights are used as the starting point to fine-tune a specific architecture for the automatic discrimination between PD and HC subjects. The fine-tuning process is performed with data from the Hw-DB corpus. Once the architecture is fine-tuned, we take the embedding of the flattened feature map as a feature vector to feed the SVM classifier.

#### 3.1.2. Dynamic Analysis

Although the image-based representation method has the advantage of being suitable for online and offline handwriting, it does not allow modeling motor problems such as rigidity, bradykinesia, or freezing of the upper limbs. These motor problems affect the dynamics of handwriting, leading to changes in velocity, acceleration, and fluency, which cannot be captured from the resulting image. In contrast, the dynamic approach leverages all the available data from online handwriting, where different feature sets are computed using the time series signals provided by the digital tablet. Table 2 shows a summary of all computed features, the details of which are described below.
***Pressure-based features:*** Pressure features describe the mechanical force exerted on the pen tip during on-surface movements produced along the writing process. Common pressure features are based on statistical functionals computed over (1) the raw pressure signal $\left(p[n]\right)$, (2) changes in pressure signal $\left(\Delta p=p[n+1]-p[n]\right)$, (3) the rate of pressure changes over time $\left(p^{\prime }\left[n\right]=\frac{\Delta p}{\Delta t}\right)$, (4) the rate of pressure variability $\left(p^{\prime \prime }\left[n\right]=\frac{\Delta^{2}p}{\Delta t^{2}}\right)$, and (5) the pressure jerk $\left(p^{\prime \prime \prime }\left[n\right]=\frac{\Delta^{3}p}{\Delta t^{3}}\right)$. Additionally, we can measure the number of changes in pressure (NCP) and the relative number of changes in pressure $\left(RNCP=\frac{NCP}{time}\right)$, using $\left(\Delta p,p^{\prime }\left[n\right],and\;p^{\prime \prime }\left[n\right]\right)$. The resulting feature set includes a total of 26 pressure features.***Kinematic features:*** According to the status of the *z*-axis signal, handwriting signals can be grouped into on-surface and in-air movements. On-surface samples correspond to digits’ strokes; conversely, in-air samples correspond to hand movements during the transition between digits. To compute kinematic features from on-surface and in-air movements, we employed the set of signals {x,y,azimuth,altitud}. Notice that features computed from the *z*-axis signal contain samples of in-air movements only. Furthermore, we employ the *x* and *y* axes to compute the pen trajectory r[n] defined in Equation (Equation 1), and the pen displacement Di[n] defined in Equation (2). Then, the set of kinematic features contains (1) movement descriptors as {Δx,1Δy,1Δazimuth,1Δaltitud,1Δz,1r[n],1Di[n]}, (2) velocity descriptors computed as changes of the previous signals over time $\left(sig_{vel}\left[n\right]=\frac{\Delta sig}{\Delta t}\right)$, (3) acceleration descriptors as changes of velocity over time $\left(sig_{acc}\left[n\right]=\frac{\Delta sig_{vel}}{\Delta t}\right)$, (4) jerk as the changes of acceleration over time $\left(sig_{jerk}\left[n\right]=\frac{\Delta sig_{acc}}{\Delta t}\right)$, and (5) number of changes of velocity and acceleration, NCV and NCA, respectively. Finally, we extracted a total of 120 and 140 kinematic features from on-surface and in-air movements, respectively. Table 2 presents a comprehensive summary of the features extracted in this work.(1)$$\begin{array}{cc} r\left[n\right] & =\sqrt{x{\left[n\right]}^{2}+y{\left[n\right]}^{2}} \end{array}$$(2)$$\begin{array}{cc} Di\left[n\right] & =\sqrt{{\left(x\left[n+1\right]-x\left[n\right]\right)}^{2}+{\left(y\left[n+1\right]-y\left[n\right]\right)}^{2}} \end{array}$$


### 3.2. Statistical Modeling

The feature computation process introduced above can result in either a scalar value or a vector (see the third column in Table 2). The resulting vectors generate a feature matrix per subject. To obtain a static vector representation per subject, we consider two statistical modeling strategies, one based on statistical functionals and the other one based on Gaussian Mixture Models (GMMs). Scalar values are concatenated with resulting statistical representations.

Dynamic feature vectors that change over time are commonly represented using statistical functionals to describe their statistical distribution. However, there exist other robust methods like the GMM that can be applied to represent dynamic phenomena. In this section, we compare classical statistical modeling based on functionals vs. GMM modeling.

#### 3.2.1. Statistical Functionals

Statistical functionals are used to describe properties of data distributions. These functionals capture specific characteristics such as central tendency, variability, or higher-order moments. This strategy is commonly used to obtain fixed representations from time- dependent signals. We consider 4 statistical fuctionals, namely, mean, standard deviation, skewness, and kurtosis, forming a static representation per feature vector. The dimensionality of the kinematic feature vector is now 4×52=208, while the dimensionality of of the pressure feature vector is 4×5=20.

#### 3.2.2. Gaussian Mixture Models (GMMs)

GMM representations enable modeling complex dynamics of time-dependent signals. GMM is a probabilistic model that combines multiple Gaussian distributions to obtain a tighter representation of the data distribution. Equation (Equation 3) defines a GMM with *M* Gaussians, where each Gaussian’s contribution is weighted by the parameter *c*. The parameters $\mu_{m}$ and $\Sigma_{m}$ represent the mean vector and covariance matrix of the *m*-th Gaussian, respectively.(3)$$f\left(x\right)=\sum_{m=1}^{M}c_{m}N\left(x;\mu_{m},\Sigma_{m}\right),$$

The idea behind the GMM model is that each Gaussian in the mixture models a sub-population along the temporal dynamics. Once the GMM is calculated, a fixed representation is obtained by combining the mean vectors and covariance matrices of all the Gaussians in the mixture. This representation is known as the GMM supervector (λ) [36]. The dimension of the supervector depends on the number of features in the input matrix and the number of Gaussians, where $\lambda_{dimension}=M\times F\times 2$. *M* is the number of Gaussian components, and *F* is the number of vector measurements in the feature matrix. We optimized the number of Gaussians based on the obtained accuracy during training. The dimensionality of the supervector that results to represent a given phenomenon or feature vector is computed as follows: optimal number of Gaussians × number of feature vectors × 2. This last number appears because the supervector is formed with the entries of the mean vector and the diagonal of the covariance matrix. For instance, when the optimal number of Gaussians is GMM with M=4, the cardinality of the resulting λ is 416-dimensional (4×52×2).

## 4. Experiments and Results

### 4.1. Experimental Setup

We conducted two experiments in this work: image-based analysis and dynamic analysis. The image-based analysis involved fine-tuning a pre-trained CNN model, described in Section 3.1.1. The dynamic analysis is based on two feature sets: pressure and kinematic. Each feature set was modeled considering two approaches for comparison purposes, statistical funtionals and GMMs. All models were trained, fine-tuned, and tested following the same partitions according to a 5-fold cross-validation strategy. The hyperparameters of the SVM classifier, namely *C*, γ, and the kernel, were optimized using a grid-search, where *C* and γ$\;\in \;\{1e^{-3},1e^{-2},\cdots ,1e^{3}\}$, and kernel∈{linear,rbf}). Notice that we evaluated two more classification methods, namely Random Forest and Gradient Boosting (GB); however, we decided to report only the ones obtained with the SVM because all accuracies with that classifier were higher.

### 4.2. Experiment 1: Image-Based Analysis

The pre-trained CNN architecture used to discriminate between non-modified and modified sequences was further used as a feature extractor for images in the Hw-DB corpus. We took the flattened output of the last convolutional layer as a feature vector to feed an SVM classifier to distinguish between PD patients and HC subjects. To improve the characterization capability of the pre-trained CNN, we considered four fine-tuning schemes over the base model using data from Hw-DB: fully frozen, partially frozen, semi-frozen, and unfrozen. In the fully frozen schema, the weights and biases of all layers in the CNN were fixed during the training process; therefore, the pre-trained CNN only used the weights and biases obtained with the synthetic data. In the partially frozen schema, the first convolutional layer is unfrozen; thus, only the weights and biases of this layer can be fine-tuned with the Hw-DB data. In the semi-frozen schema, the first and second convolutional layers were fine-tuned. Finally, in the unfrozen schema, the weights and bias of all convolutional layers were fine-tuned. Results of the image-based experiments are shown in Table 3.

Results indicate a progressive improvement in terms of accuracy when more convolutional layers are unfrozen. This suggests that the pre-trained model needs more knowledge from Parkinson’s data (i.e., the Hw-DB corpus). However, a slight decrease in the model’s accuracy is observed when the three convolutional layers are unfrozen. This could be attributed to the limited amount of data in the Hw-DB, which may not be sufficient to properly fine-tune all filters in the layers. The semi-frozen schema shows the best results, with an accuracy of up to 62.3%.

### 4.3. Experiment 2: Dynamic Analysis

Pressure and kinematics features, extracted from the time series provided by the digital tablet, are used in this experiment. An early fusion of these two feature sets was also considered. We consider two strategies to obtain a fixed representation from these time-dependent characterization strategies, namely statistical functionals and GMMs. Although the first strategy is simple, it constitutes a well-established approach to statically represent phenomena that originate from time-dependent feature sets. Concerning the GMM-based strategy, the number of Gaussians was optimized according to the accuracy obtained in training such that M=2,4,6,…,30. Figure 3 shows the results obtained in the training process with each number of Gaussians per feature set. Notice that they are sorted from the lowest to the highest accuracy. Therefore, the optimal number of Gaussians is the one indicated at the right-hand side of each figure.

Table 4 summarizes the results obtained in this experiment with both strategies (statistical functionals and GMMs). Results indicate that pressure is the feature set with the best performance, achieving an accuracy of 75% when using the statistical functionals. Notice that results based on statistical functionals outperformed those obtained with GMMs when the pressure feature set was considered. In contrast, when only kinematic features were considered, both strategies showed similar results. Finally, results with the early fusion strategy did not show improvement.

## 5. Discussion

Three different approaches were presented in this paper: one based on images resulting from the handwriting process and two based on time-dependent signals collected from a digital tablet. This discussion includes remarkable aspects of each approach and a comparison among them.

Our experiments show that classification results with the image-based features improve as the architecture adjusts more weights. The best result is obtained in the semi-frozen schema, where we allowed the fine-tuning of weights in two convolutional layers. The unfrozen schema shows slightly less accuracy. This behavior suggests that the CNN needs more real samples from the database to obtain better classification results.

The dynamic analysis yields the best results when considering features extracted from the pressure signal. These results outperform those obtained with the kinematic feature set, suggesting that vertical control of hand movement while writing is more challenging for PD patients than the horizontal ones.

When comparing models based on statistical functionals vs. GMMs, we found that the first approach is more accurate when considering the pressure feature set. Kinematic features yielded similar results with both statistical modeling strategies.

Figure 4 shows the ROC curves and their corresponding distribution of scores for the best results achieved with each modeling approach. The AUC values confirm the aforementioned claims. Image-based features are outperformed by the dynamic analyses with the pressure feature set using statistical functionals. This result is in line with the findings reported in Table 3 and Table 4, where the best accuracy obtained with the dynamic approach outperformed by up to 13% the best accuracy obtained with the image-based approach. We believe that the low performance of the imaged-based approach is because, although micrographia is a key feature in handwriting deficiencies, the writing process itself (i.e., the dynamic of handwriting over time) is the one that reveals motor symptoms more clearly.

We believe that this study constitutes a step forward in the development of automated systems that could be used to diagnose and monitor PD progression. For future work, we will focus on modeling transitions that occur while writing because our previous works show promising results [37] and because other studies have recently reported interesting insights towards this direction in neurodegenerative and immune diseases [38].

## 6. Conclusions

The present study showed that time-dependent feature sets extracted from digital tablets (namely, pressure and kinematic) are more accurate than those features extracted from images resulting from the writing process, when discriminating between PD patients and HC subjects. Additionally, our results show that the fusion of pressure and kinematic features does not improve the accuracy. Time-dependent features are represented following two strategies, one based on statistical functionals and another one based on GMMs. The comparison between using statistical functionals and using GMMs showed that the first is more suitable and yields higher accuracies. The main advantage of using statistical functionals is their direct interpretability, while the use of GMMs requires more sophisticated computations and their interpretation is not as easy, therefore limiting their use in clinical practice. This paper only considered the writing of numbers from 0 to 9; we believe that the results will stay the same when considering other tasks, like those based on words or sentences, because all these tasks are natural and belong to the set of daily living activities. Therefore, findings reported in this paper constitute a step forward in the process of creating clinically informative methods to model Parkinson’s symptoms. Further research is required considering more writing tasks and also a larger number of patients to make it possible to create more sophisticated models like those based on recurrent neural networks or long short-term memory (LSTM) models.

## Figures and Tables

**Figure 1 diagnostics-15-00381-f001:**
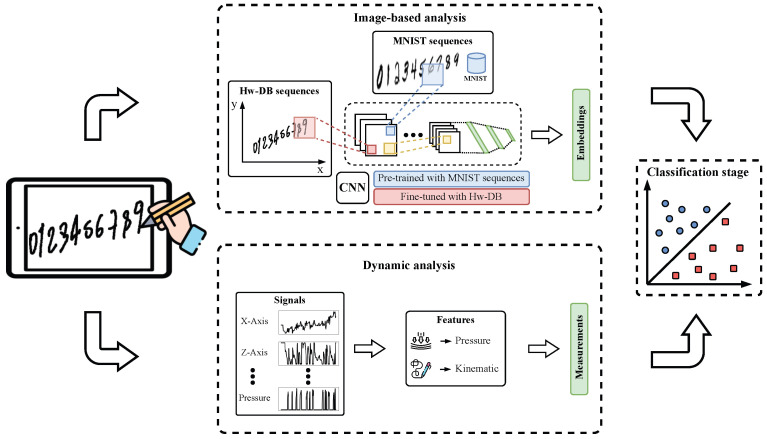
General methodology proposed in this study.

**Figure 2 diagnostics-15-00381-f002:**
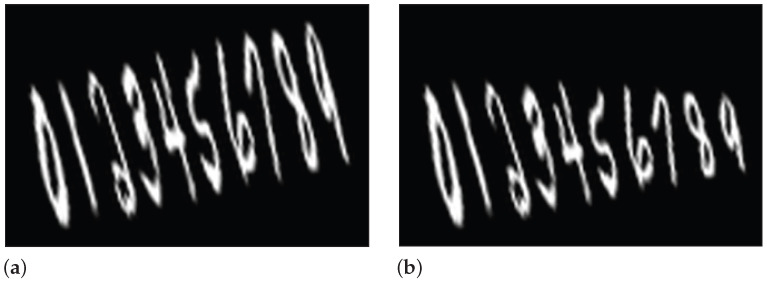
Sequences created by MNIST database. (**a**) Non-modified sequence. (**b**) Modified sequence.

**Figure 3 diagnostics-15-00381-f003:**
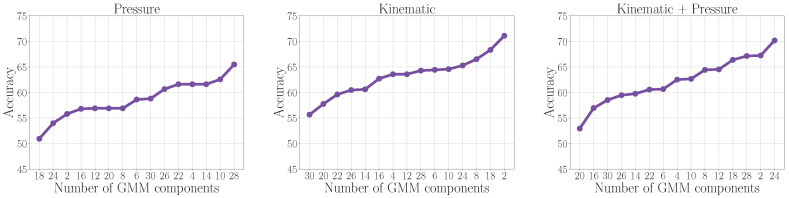
Accuracy mean obtained in training folds with different feature sets when changing the number of Gaussians in the GMM models.

**Figure 4 diagnostics-15-00381-f004:**
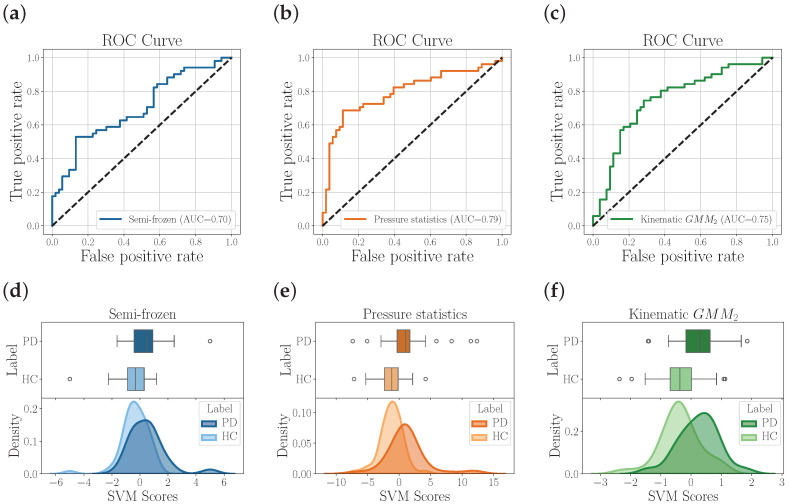
Results of the image-based analysis and the two dynamic analyses (modeled with statistical functionals and GMMs). The first row presents the ROC curves for (**a**) semi-frozen scheme, (**b**) pressure features modeled by statistical functionals, and (**c**) kinematic features modeled by two GMM components. The second row (**d**–**f**) depicts the distribution of the corresponding SVM decision scores resulting from the classification with the three feature sets.

**Table 1 diagnostics-15-00381-t001:** Demographic and clinical information of the participants.

	PD Patients		HC	
	Male	Female	Male	Female
Number of subjects	24	27	32	21
Age (μ ± σ) ⋆	69.2 ± 10.0	62.1 ± 12.0	67.1 ± 10.6	58.8 ± 10.8
Age range	50–90	29–84	49–85	43–83
MDS-UPDRS-III (μ ± σ)	39.2 ± 17.0	32.3 ± 15.9		
Range of MDS-UPDRS-III	16–82	14–77		

⋆ Indicates that there is no statistical difference between the two groups according to a *t*-test with *p*-value = 0.45. Gender bias was discarded through a Chi-squared test with *p*-value = 0.2439.

**Table 2 diagnostics-15-00381-t002:** Summary of the computed feature in each set. s = scalar value, and v = vector of elements.

Set	Feature	s/v	Description
Pressure	$p\left[n\right],\Delta p,p^{\prime }\left[n\right],p^{\prime \prime }\left[n\right],and\;p^{\prime \prime \prime }\left[n\right]$	v	Raw pressure, pressure changes, first, second and third derivatives
	NCP, and RNCP	s	Number of local extrema of pressure
	r[n],Di[n],Δx,Δy,Δazimuth,Δaltitud,Δz	v	Trajectory, displacements, and signal changes.
	Velocity	v	Velocity computed as changes in signals w.r.t. time
Kinematic	Acceleration	v	Acceleration computed as changes in signal velocity w.r.t. time
	Jerk	v	Jerk computed as changes in signal acceleration w.r.t. time
	NCV and RNCV	s	Number of local extrema for velocity
	NCA and RNCA	s	Number of local extrema for acceleration

**Table 3 diagnostics-15-00381-t003:** Results from the image-based analysis of the digit sequences using a pre-trained CNN. Values reported in terms of (μ±σ).

Fine-Tuning	Accuracy (%)	Specificity (%)	Sensitivity (%)	F1-Score (%)
Fully frozen	55.7 ± 9.3	59.6 ± 19.9	50.7 ± 13.3	0.52 ± 0.10
Partially frozen	57.6 ± 8.4	61.5 ± 19.8	52.9 ± 13.3	0.54 ± 0.09
**Semi-frozen**	62.3±11.8	65.5±18.2	58.7±18.5	0.60±0.13
Unfrozen	61.4 ± 9.4	63.6 ± 17.8	58.7 ± 13.5	0.59 ± 0.09

**Table 4 diagnostics-15-00381-t004:** Values are reported as (μ±σ).

Features	Accuracy (%)	Specificity (%)	Sensitivity (%)	F1-Score (%)
**Statistical fuctionals**				
**Pressure**	**75.0 ± 5.2**	**79.3 ± 12.1**	**70.5 ± 7.2**	**73.4 ± 5.1**
Kinematic	71.3 ± 12.4	73.8 ± 21.4	68.5 ± 15.0	69.9 ± 11.7
Kinematic + Pressure	71.3 ± 14.1	70.1 ± 19.2	72.5 ± 13.0	71.4 ± 12.7
**GMMs**				
Pressure (with 28 Gaussians)	65.5 ± 8.2	62.2 ± 15.7	68.5 ± 4.9	66.3 ± 3.3
**Kinematic with 28 Gaussians**	**71.1 ± 7.1**	**73.5 ± 15.5**	**68.4 ± 13.5**	**69.5 ± 8.3**
Kinematic + Pressure (with 28 Gaussians)	70.2 ± 15.6	69.5 ± 17.4	71.1 ± 21.8	69.5 ± 17.1

## Data Availability

The original contributions presented in this study are included in the article. Further inquiries can be directed to the corresponding author.

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
