# Peer review of "Towards Parkinson’s Disease Detection Through Analysis of Everyday Handwriting"

_diagnostics, 2025, doi:10.3390/diagnostics15030381_

Round 1

Reviewer 1 Report

Comments and Suggestions for Authors

The paper proposes a novel approach to the detection of Parkinson's disease (PD) through handwriting analysis. The authors explore two primary methods: (1) image-based analysis using a convolutional neural network (CNN) pre-trained on the MNIST dataset and fine-tuned with PD data, and (2) dynamic analysis using time-series features extracted from digital tablets during handwriting tasks. The study evaluates the effectiveness of these approaches.

My comments follow.

The authors overlooked some recent literature on machine learning based handwriting analysis for neurodegenerative disease prediction.  Below are some papers on this topic that would enrich the discussion of related work: A Machine Learning Approach to Analyze the Effects of Alzheimer’s Disease on Handwriting Through Lognormal Features (IGS 2023); From Handwriting Analysis to Alzheimer’s Disease Prediction: An Experimental Comparison of Classifier Combination Methods (IGS 2023); Deep transfer learning algorithms applied to synthetic drawing images as a tool for supporting Alzheimer’s disease prediction (MVA 2022); Dynamically enhanced static handwriting representation for Parkinson's disease detection (PRL 2019); Dynamic Handwriting Analysis for the Assessment of Neurodegenerative Diseases: A Pattern Recognition Perspective (IEEE RBE 2018); Dynamic handwriting analysis for neurodegenerative disease assessment: A literary review (App. Sci. 2019); 

The study reports relatively low accuracy (~62.3%) for the image-based approach, likely due to the limited dataset size and lack of diverse training data. I suggest the authors test alternative architectures to improve performance.

Regarding the feature extraction process, the selection of statistical functionals and GMM parameters could introduce bias. Exploring additional automated feature selection techniques could improve robustness. This should be discussed by the authors.

The study focuses only on writing digits (0-9). This probably limits the effectiveness of the proposed approach, as additional tasks such as letters, words, and sentences would allow the proposed system to achieve much better performance. However, I suspect that training a model for each individual letter (after a segmentation step) and then combining the responses would also allow the system to achieve better performance. These are points for the authors to discuss. 

Furthermore, investigating CNN hyperparameter tuning, SVM hyperparameter tuning (gamma, C, and kernel), and testing other algorithms (random forest, boosting, etc.) has the potential to improve the results obtained. 

Author Response

Dear Editor and Reviewers,

We would like to thank you for your support in the review process of the manuscript diagnostics-3428932 entitled Towards Parkinson’s Disease Detection through Analysis of Everyday Handwriting”.

We also would like to thank the reviewers for their time, constructive comments, support and dedication in reviewing the manuscript. We acknowledge that their comments helped to improve the manuscript. Please, find below the answers to the reviewer’s comments. All corrections were considered and highlighted in blue in the corrected manuscript.

REVIEWER 1

Comments and Suggestions for Authors

The paper proposes a novel approach to the detection of Parkinson's disease (PD) through handwriting analysis. The authors explore two primary methods: (1) image-based analysis using a convolutional neural network (CNN) pre-trained on the MNIST dataset and fine-tuned with PD data, and (2) dynamic analysis using time-series features extracted from digital tablets during handwriting tasks. The study evaluates the effectiveness of these approaches.

My comments follow.

The authors overlooked some recent literature on machine learning based handwriting analysis for neurodegenerative disease prediction. Below are some papers on this topic that would enrich the discussion of related work: A Machine Learning Approach to Analyze the Effects of Alzheimer’s Disease on Handwriting Through Lognormal Features (IGS 2023); From Handwriting Analysis to Alzheimer’s Disease Prediction: An Experimental Comparison of Classifier Combination Methods (IGS 2023); Deep transfer learning algorithms applied to synthetic drawing images as a tool for supporting Alzheimer’s disease prediction (MVA 2022); Dynamically enhanced static handwriting representation for Parkinson's disease detection (PRL 2019); Dynamic Handwriting Analysis for the Assessment of Neurodegenerative Diseases: A Pattern Recognition Perspective (IEEE RBE 2018); Dynamic handwriting analysis for neurodegenerative disease assessment: A literary review (App. Sci. 2019);

[Answer:] Thank you very much for your comment. The suggested references have been incorporated in the corrected manuscript.

The study reports relatively low accuracy (~62.3%) for the image-based approach, likely due to the limited dataset size and lack of diverse training data. I suggest the authors test alternative architectures to improve performance.

[Answer:] We use LeNet because it is simple and its size allows a better tuning for PD. Additionally, our preliminary experiments [ref] showed that it is very good for modeling digits, therefore we decided to start from that point and performing a fine tunning to see how far this approach could go. Now with the results reported in the manuscript we can conclude that this approach is limited compared to dynamics approaches. To improve current results it would be necessary to increase data-size. We are currently collecting more data for this purpose but to have a big-enough dataset will take a bit of time.

Note that the above mentioned reference has been added to the manuscript as [38].

[ref] J.D. Gallo-Arisitizábal, D. Escobar-Grisales, C.D. Ríos-Urrego, P.A. Pérez-Toro, E. Nöth, A. Maier, and J.R. Orozco-Arroyave, "Assessment of handwriting in patients with Parkinson's disease using non-intrusive tasks", Presented at the IEEE ISBI 2023. DOI 10.1109/ISBI53787.2023.10230617

Regarding the feature extraction process, the selection of statistical functionals and GMM parameters could introduce bias. Exploring additional automated feature selection techniques could improve robustness. This should be discussed by the authors.

[Answer:] Thank you for this suggestion. However, to do feature selection it would be necessary to have more data, which is not possible right now. The process we followed for feature extraction was always blind regarding test data, therefore we do not consider that it could introduce bias other than the one related with the data-size. Addtionally, to avoid possible biases during the optimization of the classifier, we implemented a stratified k-fold nested cross-validation strategy which guarantees minimum bias while maximizing accuracy.

The study focuses only on writing digits (0-9). This probably limits the effectiveness of the proposed approach, as additional tasks such as letters, words, and sentences would allow the proposed system to achieve much better performance. However, I suspect that training a model for each individual letter (after a segmentation step) and then combining the responses would also allow the system to achieve better performance. These are points for the authors to discuss.

[Answer:] We thank the reviewer for this comment and agree on the fact that having included other writing tasks (e.g., with letters) could have resulted on a better model; however, our study is focused on modeling handwriting during every-day activities. We considered that writing digits is more natural for patients because every human gets use to write its ID number. Additionally, working with digits results in simpler methods to perform segmentation and model training. Additionally, we wanted to extend the hypothesis previously introduced in our paper mentioned lines above [ref], in which we showed that a simple architecture is suitable to model digits.

[ref] J.D. Gallo-Arisitizábal, D. Escobar-Grisales, C.D. Ríos-Urrego, P.A. Pérez-Toro, E. Nöth, A. Maier, and J.R. Orozco-Arroyave, "Assessment of handwriting in patients with Parkinson's disease using non-intrusive tasks", Presented at the IEEE ISBI 2023. DOI 10.1109/ISBI53787.2023.10230617

Furthermore, investigating CNN hyperparameter tuning, SVM hyperparameter tuning (gamma, C, and kernel), and testing other algorithms (random forest, boosting, etc.) has the potential to improve the results obtained.

[Answer:] Thanks for pointing this out. We implemented two more classifiers and, in all cases, the results were below the ones obtained with the SVM. For the sake of clarity, we have added the following text in Section 4.1. Experimental setup:

Notice that we evaluated two more classification methods, namely Random Forest and Gradient Boosting (GB); however, we decided to report only the ones obtained with the SVM because all accuracies with that classifier were higher.

Reviewer 2 Report

Comments and Suggestions for Authors

Dear Authors,

I had the pleasure of reading your manuscript, and I found it truly interesting.

Why did the authors decide to analyze numbers instead of sentences/words? Is this supported by any previous evidence? Sentences may be more accurate than numbers from a neurological point of view.

In recent years, it has become increasingly evident that neurological signs can be observed remotely, even in the absence of a neurologist. This is particularly relevant in the context of Parkinson's Disease. I would suggest expanding on this point in the discussion section, incorporating an exploration of this concept. Additionally, the authors should reference an interesting paper that aligns with their findings (DOI: 10.1038/s41746-024-01386-0).

Author Response

Dear Editor and Reviewers,

We would like to thank you for your support in the review process of the manuscript diagnostics-3428932 entitled Towards Parkinson’s Disease Detection through Analysis of Everyday Handwriting”.

We also would like to thank the reviewers for their time, constructive comments, support and dedication in reviewing the manuscript. We acknowledge that their comments helped to improve the manuscript. Please, find below the answers to the reviewer’s comments. All corrections were considered and highlighted in blue in the corrected manuscript.

REVIEWER 2

Comments and Suggestions for Authors

Dear Authors,

I had the pleasure of reading your manuscript, and I found it truly interesting.

[Answer:] We Thank the reviewer for his/her kind comments.

Why did the authors decide to analyze numbers instead of sentences/words? Is this supported by any previous evidence? Sentences may be more accurate than numbers from a neurological point of view.

[Answer:] Our study is focused on modeling handwriting during every-day activities. We considered that writing digits is more natural for patients because every human gets use to write his/her ID number. Working with digits results in simpler methods to perform segmentation and model training. Additionally, we wanted to extend the hypothesis previously introduced in our paper mentioned lines above [ref], in which we showed that a simple architecture is suitable to model digits. Notice that this reference has been appropriately added to the corrected manuscript.

[ref] J.D. Gallo-Arisitizábal, D. Escobar-Grisales, C.D. Ríos-Urrego, P.A. Pérez-Toro, E. Nöth, A. Maier, and J.R. Orozco-Arroyave, "Assessment of handwriting in patients with Parkinson's disease using non-intrusive tasks", Presented at the IEEE ISBI 2023. DOI 10.1109/ISBI53787.2023.10230617

In recent years, it has become increasingly evident that neurological signs can be observed remotely, even in the absence of a neurologist. This is particularly relevant in the context of Parkinson's Disease. I would suggest expanding on this point in the discussion section, incorporating an exploration of this concept. Additionally, the authors should reference an interesting paper that aligns with their findings (DOI: 10.1038/s41746-024-01386-0).

[Answer:] Thanks for this suggestion. The reference and a corresponding text have been added at the end of the Discussion section.

Round 2

Reviewer 1 Report

Comments and Suggestions for Authors

The authors implemented the changes I suggested for the previous manuscript version.  However, some references need corrections, e.g.,  reference [6], where the authors' names are shown erroneously. 

Author Response

Dear Editor and Reviewers,

We apologize for the wrongly written names in reference [6]. It has been corrected in the manuscript.

 Thank you for your feedback.

Prof. Dr.-Ing. Juan Rafael Orozco-Arroyave